# Do Age-Friendly Rural Communities Affect Quality of Life? A Comparison of Perceptions from Middle-Aged and Older Adults in China

**DOI:** 10.3390/ijerph18147283

**Published:** 2021-07-07

**Authors:** Jingyu Yu, Guixia Ma, Shuxia Wang

**Affiliations:** 1School of Civil Engineering, Hefei University of Technology, Hefei 230009, China; yujingyu@foxmail.com; 2Suzhou University, Suzhou 234000, China; skxwsx@163.com

**Keywords:** age-friendly communities, older adults, quality of life, rural community

## Abstract

The aging population in rural areas of China faces serious challenges due to urban–rural disparities. In order to improve the active aging of rural older adults, the establishment of age-friendly communities is encouraged. However, globally, the focus is on age-friendly communities in urban areas, not reflecting rural communities. Hence, we addressed the importance of age-friendly rural communities (AFRCs) and aimed to investigate their impact on the quality of life (QoL) of older adults. We examined different perceptions of AFRCs among older adults (aged over 60) and middle-aged people (45–60) in rural communities with questionnaire surveys (*n* = 470 and 393, respectively). Several statistical methods, such as Chi-squared test, *t*-test, reliability test, and multiple regression, were adopted to investigate and compare the perceptions of these two. The results indicated that (1) middle-aged people were more satisfied with AFRC components and had a higher QoL than older adults; (2) the QoL of middle-aged people was predicted by housing, accessibility, and outdoor spaces; (3) the QoL of older adults was affected by housing, outdoor spaces, social participation, and public transportation. These findings aid in our understanding of rural communities and the QoL of rural residents. They are helpful for urban planners and policymakers to improve the planning of AFRCs and supplement research on age-friendly communities in rural areas. Practical implementations are proposed for the planning of AFRCs, such as the passive design of residential housing, grouping of community facilities together, and improvement in the hygiene of outdoor spaces in rural areas.

## 1. Introduction

In 2019, there were 176 million older adults aged 65 and over in China, accounting for 12.6% of the total population [1]. The speed of population aging is rapidly increasing and turning China into an aged society, which will occur when the proportion of older adults surpasses 14% [2]. More importantly, the proportion of the elderly population is much higher in rural areas than in urban areas. Alongside the processes of industrialization and urbanization seen in China, there is a mass migration of labor from rural areas to cities as people seek money-making jobs [3]. The goal was set for 100 million migrant rural labors to settle down in cities by 2020 [4]. Young adults’ migration to cities have led to a rapid rise in the proportion of older adults in rural communities.

Compared with urban areas, the phenomenon of population aging has become more challenging in rural areas. Due to backward economic development, per capita disposable income in rural areas (i.e., 16,021 Renminbi (RMB) in 2019) is much lower than in urban areas (i.e., 42,359 RMB in 2019) [1]. This exacerbates the dilemma of “getting old before getting rich” in rural areas, where the supply of community services is deficient, with only a 45.3% coverage rate, while the coverage rate is 78.7% in urban areas [5]. Limited incomes and community support leave rural older adults more vulnerable when their children migrate to cities. In order to actively respond to the aging rural population, the government has invested in rural healthcare services and encouraged aging in place for rural older adults [6]. The rural revitalization program, published in 2018, also proposed the development of a multilevel rural elderly care system based on home care, supported by communities and supplemented by senior residential and nursing institutions [7]. Aging-friendly and age-livable communities are essential to support active aging and improve well-being and health among rural community-dwelling older adults [8]. Therefore, age-friendly communities (AFCs) are encouraged by the government to provide a supportive and convenient living environment for older people [9]. The Chinese government published several guidelines and regulations to promote AFC. Although their importance is widely recognized, relevant planning and design standards are still lacking, especially for rural areas. Studies on age-friendly rural communities (AFRCs) remain rare and mainly focus on Canada and Australia [10,11]. Given the huge demand for, and importance of, AFRCs in China, it is surprising that there is so little research on the impact of AFRCs on older adults.

As noted above, due to the large-scale labor migration, the proportion of middle-aged and aged people in the Chinese rural population has increased. According to the population census in 2010, the middle-aged population (aged 45–59) in rural areas was 168 million, accounting for 20.4% of the total rural population, and the proportion of the aging population (60 and above) in rural areas was 15.0% [12]. The middle-aged are healthier than older adults, have fewer children because of the birth-control policy and have a higher education level because of the popularization of compulsory education. Because of the discrepancy in sociodemographic characteristics between middle-aged and older people, there might be generational distinctions in their perception of AFRCs. The middle-aged will turn into the aging population and have fewer migration opportunities. Their perceptions of AFRCs are valuable for policymaking and urban planning. Hence, this study aims not only to investigate the effect of AFRCs on the quality of life (QoL) of older adults, but also to compare perceptions of AFRCs between middle-aged and older adults. 

## 2. Literature Review

In order to support active aging and aging in place, the concept of AFC has been proposed to make the living environment of older adults more age-friendly and livable [13]. AFCs are being established all over the world, such as the Global Age-Friendly Cities Project [14], AFCs in the United States and Canada [15,16], age-friendly environments in Europe [17], and age-friendly practices in East Asia [18]. AFCs, which are characterized by sufficient outdoor spaces and public transportation, an accessible living environment, and convenient community services, support older adults in maintaining physical activities [19], improve well-being and life satisfaction [20], and reduce social isolation [21]. However, AFCs stemmed from the concept of age-friendly cities, with an emphasis on making the urban environment age-friendly, not the rural environment [11].

The rural aging population encounters additional challenges in their communities, such as limited infrastructure, limited transportation networks, and lack of support from community healthcare services [22]. The eight characteristics of age-friendly cities, identified by the World Health Organization (WHO), are not a good fit for rural communities. Policymakers and researchers from Canada and Australia have made great efforts to conceptualize and develop the framework of AFRCs. Based on the WHO framework of age-friendly cities, the Canadian Age-Friendly Rural and Remote Communities project identified eight features of AFRCs by collecting data from ten rural and remote communities [23]. All these AFRC features are specific to rural settings. Keating et al. conceptualized the AFRC using mixed methods. They argued that AFRCs should be more inclusive, interactive, and dynamic to suit the community needs and resources [23]. Hancock et al. investigated the perceptions of rural community-dwelling older adults of AFC based on a study in Australia [10]. They proposed key issues and concerns relevant to AFRCs, such as the need for equity in service provision, greater choice in community services, and more transport options. Brooks-Cleator and Giles suggested collaborating with older adults to develop territory-wide, age-friendly rural and remote communities in order to support the physical activity of older adults in rural areas [24]. There is also abundant discussion of public transportation, participation in community activities, and community services for rural-community-dwelling older adults [25,26].

AFRCs have not only physical features, such as housing, accessibility, public transportation, and outdoor spaces, but also social features, including social participation, community services, and communication and information [27]. To age well in rural areas, it is essential to have housing that provides safe and comfortable shelter that meets the living and agricultural needs of older adults. The interior spaces and level surfaces need to be non-slip and well-maintained to ensure the safety of senior residents [27]. For rural-community-dwelling older adults, the outdoor spaces are sufficient, in contrast to urban areas. However, the provision of public facilities in rural communities, such as toilets and footpaths, needs to be well-planned [10]. Rural-community-dwelling older adults need reliable, affordable, and frequent public transport to all rural areas and services [28]. Community services, including daycare, healthcare, and medical services, are accessible to older adults living in rural areas in order to meet their basic needs. The pavements are non-slip and barrier-free because the majority of older adults seldom drive and depend on their walking ability. Therefore, public transport stops and stations need to be installed within walking distance of older adults in rural areas [29].

Social participation activities help older adults to maintain social connections and receive social support from communities [30]. Community security and community health support activities are key concerns of older adults in rural areas. The Chinese government established a primary healthcare system in both urban and rural areas. However, the communes in rural communities are owned and financed by the local government, as are most healthcare and community service facilities, which are often ineffectively run [31]. Rural older adults heavily depend on informal support from friends and family due to the lack of formal community services. Older adults in good health are encouraged to actively help their relatives and friends in rural communities [32]. Older adults also need to communicate with others and obtain accurate information. In rural communities of China, broadcasts and billboards were major approaches to deliver information. Communications systems (e.g., broadcasts and billboards) should be able to reach community residents of all ages in order to provide up-to-date and available information. Important notifications and messages are posted on centrally located billboards for rural-community-dwelling residents to easily access necessary information [33].

As the living environment has a continuous impact on older people’s daily life, AFRCs aim to ensure the health, well-being, life satisfaction, and QoL of rural older adults [13]. QoL refers to subjective assessments of the individual’s life satisfaction and well-being [33]. As a wide-ranging concept, QoL has multiple dimensions, such as personal physical health, mental health, independence, social relationships, personal beliefs, and environment [34]. QoL has also been measured on multiple domains, including self-perception, social relations, health, and community [35]. Another study evaluated the QoL of older adults in the physical, psychological, social, and environmental domains, and indicated the impact of the management of public housing facilities on different domains of QoL [36]. QoL can also be narrowed down to subjective feelings about health, illness, and treatment, which is classified as health-related QoL [37]. In view of long-term care systems, social-care-related QoL was proposed by van Leeuwen et al. to investigate the relationships between the QoL of older adults and the age-friendly environment (e.g., accessibility, information, and living environment) [38].

Previous studies have paid attention to AFCs in urban areas and their effect on the life satisfaction and QoL of older adults [39]. Limited research has been conducted on the association between AFRCs and the QoL of older adults, especially in China, an East Asian developing country with a huge aging population and unique social and economic characteristics. Based on the above literature review on AFCs and the QoL of older adults, a positive impact of AFRCs on different domains of older people’s QoL is hypothesized. Moreover, previous studies have indicated that the impact of residential environment on the mental health and physical behaviors of middle-aged and older adults was different [40,41]. Hence, the relationships between AFRCs and the QoL of older adults are hypothesized to be different for middle-aged people and older people, because of their different sociodemographic characteristics.

## 3. Research Methods

In order to investigate the impact of AFRC components on QoL and compare the perceptions of middle-aged people and older adults, a questionnaire survey was conducted among rural-community-dwelling adults. All respondents were guaranteed that their personal information and responses would remain confidential. The questionnaire surveys were completed with face-to-face assistance from trained research assistants in order to avoid any misunderstandings. Purposive sampling was applied to select appropriate respondents who: (1) were over 45 years of age at the time they took the survey; (2) had lived in their current rural community for more than six months; (3) had sufficient cognitive and linguistic abilities to understand and respond to the questionnaire. The surveys were distributed in 10 rural communities of different regions in China in order to mix adults from different rural areas. The total aging population in these study regions is near 3,000,000. According to sample size calculation methods, the sample size needs to achieve 385 or more in order to have a confidence level of 95%. Ultimately, 863 valid questionnaires were collected. Of these, 393 respondents were in the 45–60 age range, which was designated as representing middle-aged people; 470 respondents were aged over 60 and treated as older adults [37,42]. According to WHO, people aged 60 year and older are counted as the aging population [43]. Hence, respondents over 60 years old are classified as older adults. 

The questionnaire survey was designed with three major sections: (1) background information (demographics) of the respondents (age, education, income, marriage, etc.); (2) the respondents’ agreement with components of AFRCs; (3) QoL level. A five-point Likert-type scale ranging from 1 (strongly disagree/very dissatisfied) to 5 (strongly agree/very satisfied) was used to measure the respondents’ rating of community environment items and their QoL. Both AFRC components and QoL factors were measured by commonly used and validated scales. We measured the AFRC components assessing housing, accessibility, outdoor spaces, social participation, communication and information, and public transportation [14,44]. Using scales developed by the WHO [35,45], QoL factors covered several domains (physical, psychological, social, and environmental). QoL scales were adopted to the WHOQoL Chinese version in order to avoid misunderstandings with older adults.

Multiple statistical methods were adopted, including a Chi-squared test, reliability test, *t*-test, and linear regression models. The analysis was conducted using SPSS version 24.0. Preliminary analyses were first performed using a Chi-squared test to compare the demographic characteristics of middle-aged and older adults. A reliability analysis was used to examine the internal consistency of AFRC components and QoL factors. A *t*-test was then used to investigate the significant differences in the perception of AFRC components between middle-aged and older adults. Multiple linear regression models were developed to investigate the different effects of AFRC components on the QoL of middle-aged and older adults.

## 4. Results

### 4.1. The Demographic Characteristics of Middle-Aged and Older Adults

Overall, 54.4% of the respondents were older adults, aged over 60, and 45.5% were middle-aged, as shown in Table 1. The gender distribution was close to balanced, with 54.0% male and 46.0% female. Only 2.4% had attended college; most were not well-educated, with less than a high school education. In terms of living situation, 49 respondents were living alone (5.8%), 294 with their spouse (35.0%), 395 with their children (47.1%), and 101 in an intergenerational household with spouse and children (12.0%). Nearly half of the respondents (48.7%) were self-reported to be in good health; only 10.7% reported being unhealthy. Most of the respondents (92.8%) lived independently in rural areas, and 7.2% were dependent on the support of their families, friends, and healthcare services in their daily life. Over half of the respondents (61.9%) had a monthly income of less than 2500 RMB (i.e., <$386.8); only 14.9% had a monthly income of over 5000 RMB (i.e., >$773.5). 

The Chi-squared test results indicate that older adults were more likely to be male (*p* = 0.000). The older respondents were less educated than their middle-aged counterparts (*p* = 0.000). Many older adults lived with their spouse, while the middle-aged tended to be living with and raising their children (*p* = 0.000). The older adults had a lower monthly income than the middle-aged (*p* = 0.002). There were no significant differences in terms of health condition and independence between older and middle-aged respondents. These results reflect a general phenomenon of rural mobility in the process of urbanization in China. There are many young adults moving from the countryside to cities, seeking better living conditions, which means that most rural community-dwelling older adults live without their children and earn less. 

### 4.2. Reliability Analysis

In order to test the internal consistency of all factors, a reliability analysis was conducted. Six AFRC components were identified: housing (A1), accessibility (A2), outdoor spaces (A3), social participation (A4), communication and information (A5), and public transportation (A6). Four QoL factors were identified: physical QoL (Q1), psychological QoL (Q2), environmental QoL (Q3), and social QoL (Q4). As shown in Table 2, the Cronbach’s alpha of all identified factors was over 0.7, which showed that the factors were reliable, internally consistent, and valid for further analysis [45].

### 4.3. Independent Sample t-Test

An independent sample *t*-test was conducted to compare middle-aged and older residents’ satisfaction with AFRC components and QoL factors. The results in Table 3 indicated that middle-aged people were more satisfied with their housing (A1), accessibility (A2), outdoor spaces (A3), social participation (A4), communication and information (A5), and public transportation (A6). Older adults reported a lower level of QoL than their middle-aged counterparts, aside from psychological QoL (Q2).

### 4.4. Multiple Regression

A multiple linear regression analysis was applied to predict and compare the different effects of AFRC components on the QoL factors of middle-aged and older adults. As illustrated in Table 4, the results of the regression models for older adults stated that: (1) both the physical and psychological QoL (Q1 and Q2) of older adults were predicted by housing (A1) and public transportation (A6); (2) social QoL (Q3) of older adults was influenced by outdoor spaces (A3) and public transportation (A6); (3) environmental QoL (Q4) of older adults was affected by housing (A1) and social participation (A4).

The regression models in Table 5 were used to examine the contribution of AFRC components to the prediction of the QoL factors of middle-aged people. In contrast to the older adults, the results indicated that: (1) physical QoL (Q1) of middle-aged people was predicted by housing (A1) and outdoor spaces (A3); (2) psychological QoL (Q2) of middle-aged people was influenced by housing (A1) and accessibility (A2); (3) social QoL (Q3) of middle-aged people was only influenced by accessibility (A2); (4) environment QoL (Q4) of middle-aged people was impacted by accessibility (A2) and outdoor spaces (A3).

## 5. Discussion

According to the results of the independent *t*-test, middle-aged and older adults have different perceptions of AFRC components and QoL factors. Middle-aged people tend to be more satisfied with their AFRC components and QoL than older adults. Older adults in rural areas are less educated and earn less than middle-aged people, which might result in lower levels of QoL. The low QoL of older adults is consistent with previous research [46]. The daily life of older adults relies heavily on AFRC components. When AFRC components are deficient, older adults are more vulnerable than middle-aged people. Hence, older adults might report lower levels of satisfaction with AFRC components.

Figure 1 illustrates the different relationships between AFRC components and the QoL factors of middle-aged and older adults, shown by the regression results in Table 4 and Table 5. Among all AFRC components, only housing exerts the same impact on both the physical and psychological QoL of both middle-aged and older adults, which is consistent with previous studies [47]. Housing is one of the key factors improving the quality of life and health of older adults. In China, the utilization of housing land has changed greatly due to industrialization and urban development. Rural housing not only fulfills the basic living needs of residents, but also increases their income by developing a non-agricultural economy including catering, homestay hotels, and tourism [48]. Housing is, therefore, an important AFRC component, which significantly influences the QoL of rural residents in all age groups. Housing is also found to impact the social QoL of older adults. This might be attributed to the fact that older adults tend to meet their friends and maintain social connections within their own housing units.

Accessibility is found to be a key AFRC component influencing the QoL of middle-aged people, but it exerts no impact on the QoL of older adults. In China, community services and healthcare service facilities are often located in counties to support multiple rural communities in the area, but are not constructed in each rural community. People therefore need to drive or take public transportation to access community services. However, older adults often rely on their walking ability to move around in the rural community, and seldom use public transportation [49]. Hence, the accessibility of community services has no direct impact on the QoL of older adults. Middle-aged people need to access community services to deal with affairs of daily life, including financial services, business support, education of children, their parents’ medical services, etc., which affects their satisfaction with QoL.

Outdoor spaces were found to influence the environmental QoL of older adults, and affect the physical and social QoL of middle-aged people, which is consistent with previous studies [50]. Sufficient outdoor space provides older adults with access to fresh air and a feeling of comfort in rural communities [51]. Outdoor spaces are part of the living environment of rural communities, and are essential to maintain older adults’ health and satisfaction with environmental QoL. In China, there is a plan to prevent pollution and improve sewage treatment in rural areas, which will help tremendously with keeping outdoor spaces clean. Clean and convenient outdoor spaces are good for the hygiene of rural communities and benefit the physical health of community-dwelling residents (i.e., physical QoL). Middle-aged people also depend on outdoor spaces to maintain their friendships and social connections, which reflect the social domain of QoL.

Social participation might influence the social QoL of older adults. Previous research indicated that older adults in rural communities possess strong community connections and family networks [52]. Mutual help among older adults is widespread in East Asian countries, such as Japan, China, and Thailand [31,53]. In China, a community-help program was introduced, encouraging older adults to mutually help and support each other in order to avoid social isolation and improve their well-being. Hence, there are abundant activities and events for rural community-dwelling older adults to perform or attend together, such as paying bills, shopping, and farm work. Rural communities provide support to older adults, including daily care, healthcare, social services, etc. Community-based care and support allow older adults to participate in community activities, which increase the social gathering and social connection of older adults. This form of social gathering has advantages in maintaining the social relationships of older adults (i.e., social QoL).

Public transportation is a key AFRC component, which correlates positively with the QoL of older adults. Older adults rely on public transportation, because they seldom drive and community services are often far from rural communities [54]. Frequent public transportation is valuable for older adults to access public community facilities, including hospitals, banks, and police stations. It is good to maintain the physical, psychological, and environmental QoL of older adults. On the other hand, middle-aged people often drive to access community facilities. They are unaffected by the provision of public transportation.

It is remarkable to note that communication and information has no effect on the QoL of either older adults or middle-aged people. This might be attributed to the indirect impact of information on the QoL of rural-community-dwelling residents through other AFRC components. In China, the revolution of information and communication technologies has dramatically changed the lifestyle of older adults. However, the development of information technologies lags behind in rural areas, which might mean that communication and information has a weaker effect on the QoL of rural residents.

## 6. Recommendations

The results indicate that middle-aged and older adults have different perceptions of the impact of AFRC components on QoL. As middle-aged people will, in time, become older adults, their perceptions and opinions are also valuable for the development of age-friendly environments in rural areas. In order to enhance the QoL of rural community-dwelling residents, well-constructed housing should be passively designed, considering the utilization of natural ventilation and daylight. It is suggested that community facilities should be located close together, and their accessibility using public transportation should be improved. As outdoor spaces are important for the QoL of both middle-aged and older adults, it is recommended that they are kept clean, and that sufficient public facilities and toilets are installed. Public facilities, such as physical exercising facilities and billboards, should be installed at the center of rural communities, so that they are easily accessible for older adults. Residents with the capacity to work are encouraged to actively participate in rural community activities to provide support to older adults when needed.

The direct impact of AFRC components on the QoL of both middle-aged and older adults is confirmed by the current study. However, some *R*^2^ values of the regression models were relatively low. Other sociodemographic factors (e.g., income, health condition, age) perhaps influence the well-being and QoL of rural residents. Hence, further study is recommended to investigate the moderating and mediating effects of sociodemographic characteristics on the causal relationships between AFRC components and the QoL of older adults. Purposive sampling method might induce bias in the results. Therefore, it is suggested that a random sample is adopted in further studies. Moreover, it could be useful for future qualitative case studies to cross-check and validate our findings, and to deepen our understanding of the complicated interactions between AFRC components and the QoL of rural residents.

## 7. Conclusions

The aging population is rapidly increasing in rural areas of China due to the dramatic urbanization in the country. The living environment of rural communities has an impact on the health and well-being of older adults. This study investigated the direct effect of age-friendly rural communities on the quality of life of older adults and compared middle-aged and older adults’ perceptions of AFRC components. Six AFRC components (i.e., housing, accessibility, outdoor spaces, social participation, communication and information, and public transportation) and four QoL factors (i.e., physical, psychological, environmental, and social QoL) were identified. Middle-aged people reported relatively higher satisfaction with AFRC components and QoL factors than older adults. The results of regression models indicated that: (1) older adults’ QoL was affected by housing, outdoor spaces, communication and information, and public transportation; (2) the QoL of middle-aged people was influenced by housing, accessibility, and outdoor spaces.

Based on the current results, several practical implementations were proposed in order to improve the planning of AFRCs, such as the passive design of residential housing, the placement of community facilities together, and improvements in the hygiene of outdoor spaces in rural areas. It is recommended that the moderating and mediating impacts of sociodemographic characteristics on older adults’ QoL be investigated further, and that qualitative studies be conducted to cross-check the validity of this study’s findings.

## Figures and Tables

**Figure 1 ijerph-18-07283-f001:**
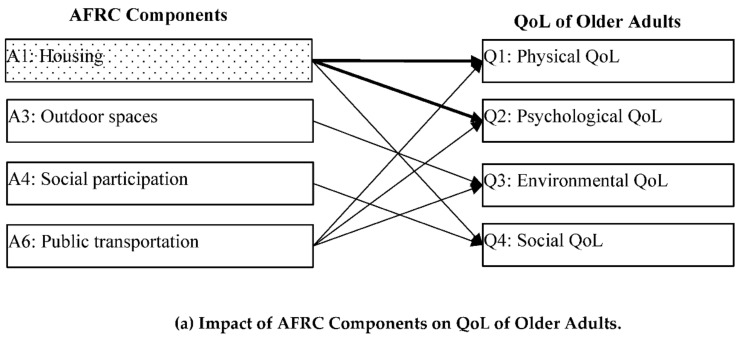
Impact of Age-friendly Rural Community (AFRC) Components on Quality of Life (QoL) of Middle-Aged and Older Adults. Note: 
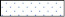
 Indicates components that significantly influence the QoL of both middle-aged and older adults.

**Table 1 ijerph-18-07283-t001:** Demographic Features of Middle-Aged and Older Adults in Rural Communities.

Variables	Total	Older Adults (%)	Middle-Aged (%)
Overall	863	470 (54.5)	393 (45.5)
Gender			
Male	459 (54.0)	306 (36.1)	153 (17.9)
Female	396 (46.0)	158 (18.4)	238 (27.6)
*p*-Value	0.000		
Education			
Primary	337 (40.4)	256 (30.3)	85 (10.1)
Middle school	288 (34.0)	123 (14.5)	165 (19.5)
High school	197 (23.2)	68 (8.0)	129 (15.2)
College or above	20 (2.4)	11 (1.3)	9 (1.1)
*p*-Value	0.000		
Living Methods			
Living alone	49 (5.8)	27 (3.2)	22 (2.6)
Living with spouse	294 (35.0)	217 (25.9)	77 (9.2)
Living with children	395 (47.1)	169 (20.1)	226 (26.9)
Living with spouse and children	101(12.0)	43 (5.1)	58 (6.9)
*p*-Value	0.000		
Health Conditions			
Good health	431 (48.7)	217 (25.6)	196 (23.1)
Neutral	344 (40.6)	176 (20.8)	168 (19.8)
Bad health	91 (10.7)	65 (7.6)	26 (3.1)
*p*-Value	0.063		
Independence			
Independent	743 (92.8)	398 (49.7)	345 (43.1)
Dependent	58 (7.2)	26 (3.8)	22 (3.4)
*p*-Value	0.680		
Income			
<1000 Reminbi (RMB) ($154.7)	163 (19.6)	113 (13.5)	51(6.1)
1000–2500 RMB ($154.7–$386.8)	353 (42.3)	172 (20.6)	181 (21.7)
2500–5000 RMB ($386.8–$773.5)	191 (22.9)	103 (12.3)	88 (10.5)
>5000 RMB ($773.5)	124 (14.9)	58 (6.9)	66 (7.9)
*p*-Value	0.002		

**Table 2 ijerph-18-07283-t002:** Reliability Analysis of AFRC Components and QoL Factors.

Factors	Items	Descriptions	Alpha (α)
Age-Friendly Rural Communities			
A1: Housing	a1	Well-constructed housing provides safe and comfortable shelter.	0.876
	a2	Interior spaces and level surfaces allow easy movement.	
	a3	The daylighting and ventilation of housing is good.	
A2: Accessibility	a4	Community services are situated together and are accessible.	0.836
	a5	Transport stations are accessible and have adequate seating and shelter.	
	a6	Community services are conveniently located and accessible by public transportation.	
	a7	Health service facilities such as hospitals are fully accessible.	
	a8	Pavements are non-slip and barrier-free for walking.	
A3: Outdoor spaces	a9	Public areas are clean and pleasant in rural communities.	0.712
	a10	Outdoor spaces in rural communities are sufficient for gathering and social activities.	
	a11	Public toilets in rural areas are sufficient, well-maintained, and accessible.	
	a12	Rural public facilities are sufficient and well-maintained.	
A4: Social participation	a13	Adequate community support services are offered for promoting quality of life of residents in rural areas.	0.794
	a14	Community activities and events can be attended by adults of different generations.	
	a15	Community activities are affordable, with no additional participation costs.	
	a16	The security and safety of all residents in rural communities can be guaranteed.	
	a17	Community service facilities are safely constructed.	
A5: Communication and information	a18	A basic and effective communication system reaches rural community residents of all ages.	0.851
	a19	Regular and widespread distribution of information is provided.	
	a20	Telephone answering services give instructions slowly and clearly for rural community residents of all ages.	
	a21	Rural residents are regularly consulted on how they can be better served.	
A6: Public transportation	a22	Public transportation costs are consistent, clearly displayed, and affordable.	0.880
	a23	All rural areas and services are accessible by public transport, with good connections and well-marked routes and vehicles.	
	a24	Public transportation services in rural communities are reliable and frequent.	
	a25	Vehicles are clean and have priority seating.	
Quality of Life (QoL)			
Q1: Physical QoL	q1	To what extent do you feel that pain prevents you from doing what you need to do?	0.708
	q2	How much do you need any medical treatment to function in your daily life?	
	q3	How well are you able to get around?	
	q4	Do you have enough energy for everyday life?	
	q5	How satisfied are you with your ability to perform your daily living activities?	
	q6	How satisfied are you with your work capacity?	
	q7	How satisfied are you with your sleep?	
Q2: Psychological QoL	q8	How satisfied are you with yourself?	0.715
	q9	How much do you enjoy life?	
	q10	How well are you able to concentrate?	
	q11	How often do you have negative feelings?	
	q12	Are you able to accept your bodily appearance?	
Q3: Environmental QoL	q13	How safe do you feel in your daily life?	0.837
	q14	How healthy is your physical environment?	
	q15	Have you enough money to meet your needs?	
	q16	How accessible is the information you need in day-to-day life?	
	q17	How satisfied are you with the conditions of your living place?	
	q18	How satisfied are you with access to health services?	
	q19	How satisfied are you with your transport?	
Q4: Social QoL	q20	To what extent do you have the opportunity for leisure activities?	0.761
	q21	How satisfied are you with your personal relationships?	
	q22	How satisfied are you with the support you get from your friends?	

**Table 3 ijerph-18-07283-t003:** Independent *t*-test Results of AFRC Components and QoL Factors.

Factors	Middle-Aged	Older Adults	Mean Difference	*t*-Value	*p*-Value
A1: Housing	3.916	3.495	0.421	5.306	0.000
A2: Accessibility	3.545	2.859	0.686	9.210	0.000
A3: Outdoor spaces	3.593	2.927	0.667	7.785	0.000
A4: Community services	3.472	2.851	0.621	7.099	0.000
A5: Communication & info.	3.475	2.767	0.708	8.484	0.000
A6: Public transportation	3.451	2.878	0.573	6.656	0.000
Q1: Physical QoL	3.605	3.404	0.201	3.803	0.000
Q2: Psychological QoL	3.713	3.721	−0.008	−0.135	0.891
Q3: Social QoL	3.738	3.447	0.292	5.912	0.000
Q4: Environmental QoL	3.833	3.599	0.234	4.127	0.000

**Table 4 ijerph-18-07283-t004:** Multiple Linear Regression of AFRC Components and QoL Factors for Older Adults.

Dependent	Independent	B	*t*	Sig. t	*R* ^2^	F	Significance. F
Q1: Physical QoL	Constant	2.834	20.882	0.000	0.109	18.427	0.000
	A6: Public transportation	0.110	3.386	0.000			
	A1: Housing	0.097	2.758	0.000			
Q2: Psychological QoL	Constant	2.831	17.180	0.000	0.104	15.614	0.000
	A1: Housing	0.143	3.322	0.001			
	A6: Public transportation	0.091	2.273	0.024			
Q3: Social QoL	Constant	2.611	22.228	0.000	0.241	54.193	0.000
	A3: Outdoor spaces	0.191	4.046	0.000			
	A6: Public transportation	0.125	2.523	0.012			
Q4: Environmental QoL	Constant	2.615	16.412	0.000	0.188	40.232	0.000
	A4: Social participation	0.256	6.430	0.000			
	A1: Housing	0.084	1.981	0.048			

**Table 5 ijerph-18-07283-t005:** Multiple Linear Regression for AFRC Components and QoL Factors for Middle-aged People.

Dependent	Independent	B	*t*	Sig.t	*R* ^2^	F	Sig.F
Q1: Physical QoL	Constant	2.351	14.720	0.000	0.153	25.593	0.000
	A3: Outdoor spaces	0.175	4.412	0.000			
	A1: Housing	0.155	3.549	0.000			
Q2: Psychological QoL	Constant	2.487	12.775	0.000	0.128	20.909	0.000
	A2: Accessibility	0.225	3.907	0.000			
	A1: Housing	0.165	3.850	0.000			
Q3: Social QoL	Constant	2.338	17.851	0.000	0.205	74.526	0.000
	A2: Accessibility	0.379	8.633	0.000			
Q4: Environmental QoL	Constant	2.719	16.609	0.000	0.119	20.042	0.000
	A3: Outdoor spaces	0.167	3.796	0.000			
	A2: Accessibility	0.137	2.108	0.036			

## Data Availability

Not applicable.

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
