# Peer review of "Do Age-Friendly Rural Communities Affect Quality of Life? A Comparison of Perceptions from Middle-Aged and Older Adults in China"

_ijerph, 2021, doi:10.3390/ijerph18147283_

Round 1

Reviewer 1 Report

Thank you for inviting me to review this manuscript on the impact of age-friendly communities on quality of life for middle aged and older adults in China. I read this manuscript with interest and generally speaking it is interesting, strong and well written. There are, however, some areas that could be improved.

Introduction

Why is the society deemed as ‘aged’ when older population surpasses 14% Does this need a reference to support this cut off?

I would avoid using language like “has mushroomed” – seems a bit casual for an academic journal.

“The middle-aged will join the aged cohort and have fewer migration opportunities in a dozen years” – I didn’t understand what this meant, particularly the reference to a dozen years.

Throughout the manuscript information and communication are mentioned as factors – but it is not clear what this refers to. What is meant by information and communication in this context? This needs clarification throughout.

Research Methods

Why are the over 60s designated or labelled as older adults? Is there a reference for this, otherwise it seems quite arbitrary.

Can the QOL tools be named in the text? And are they validated for use in a Chinese context?

Results

References to RMB for income are useful, but do they need contextualising? Would £ / $ equivalents help? Or at least cited against the Chinese relevant national / regional average?

Discussion

It is interesting that housing a key predictor of QOL – but discussion says this is inconsistent with previous research. I disagree with this, there is a wealth of evidence and existing literature that housing impacts on QOL. If this statements relates to china alone then this needs clarifying, but again I’d be surprised if there was no evidence for this in Chinese context.

Some of the statements are very forthright and need context and toning down, such as “social participation only influences the social QOL of older adults.” – in this study, which used purposive sampling. There will likely be some groups in middle aged people for who social participation influences of social QOL.

Recommendations

“Public facilities should be installed at the center of rural communities” – such as what?

Limitations are well considered- but the authors should also add this was not a randomised sample, and there may have been bias in recruitment and therefore results. That seems like a major limitation.

Author Response

Introduction

  1. Why is the society deemed as ‘aged’ when older population surpasses 14% Does this need a reference to support this cut off?

Reply:

Please kindly note that one reference has been included. Revised ‘Introduction’ section (see page 1)

“The speed of population aging is rapidly increasing and turning China into an aged society, which will happen when the proportion of older adults surpasses 14% 2.”

  1. I would avoid using language like “has mushroomed” – seems a bit casual for an academic journal.

Reply:

Revised ‘Introduction’ section (see page 2):

“As noted above, due to the large-scale labor migration, the proportion of middle-aged and aged people in the Chinese rural population has increased.”

  1. “The middle-aged will join the aged cohort and have fewer migration opportunities in a dozen years” – I didn’t understand what this meant, particularly the reference to a dozen years.

Revised ‘Introduction’ section (see page 2):

“The middle-aged will turn into aging population and have fewer migration opportunities.”

  1. Throughout the manuscript information and communication are mentioned as factors – but it is not clear what this refers to. What is meant by information and communication in this context? This needs clarification throughout.

Please kindly note that the factor named as ‘communication and information’ was indicated by WHO aging-friendly cities checklist. The detail scales of communication and information were listed in Table 2.

Revised ‘Literature Review’ section (see page 3)

“AFRCs have not only physical features, such as housing, accessibility, public trans-portation, and outdoor spaces, but also social features including social participation, community services, and communication and information 27.”

“Communications systems should be available to reach community residents of all ages in order to provide up-to-date and available information.  Centrally located and easily ac-cessible information is also important for rural community-dwelling residents32.”

Revised ‘Research Methods’ section (see page 4)

“Both AFRC components and QoL factors were measured by commonly used and validat-ed scales.  We measured the AFRC components assessing housing, accessibility, outdoor spaces, social participation, communication and information, and public transporta-tion13,42.”

Research Methods

  1. Why are the over 60s designated or labelled as older adults? Is there a reference for this, otherwise it seems quite arbitrary.

Reply:

Please kindly note that one reference has been included. Revised ‘Research Methods’ section (see page 4)

“Ultimately, 863 valid questionnaires were collected.  Of these, 393 respondents were in the 45–60 age range, which was designated as representing middle-aged people; 470 re-spondents were aged over 60 and treated as older adults 37, 43.”

  1. Can the QOL tools be named in the text? And are they validated for use in a Chinese context?

Reply:

Revised ‘Research Methods’ section (see page 4)

 “Both AFRC components and QoL factors were measured by commonly used and validat-ed scales.  We measured the AFRC components assessing housing, accessibility, outdoor spaces, social participation, communication and information, and public transporta-tion13,42.  Using scales developed by the WHO35,44, QoL factors covered several domains (physical, psychological, social, and environmental). QoL scales was adopted the WHOQoL Chinese version in order to avoid misunderstanding of older adults.”

Results

  1. References to RMB for income are useful, but do they need contextualising? Would £ / $ equivalents help? Or at least cited against the Chinese relevant national / regional average?

 Reply:

Please kindly note that income has been calculated with US dollars. Table 1 has been revised (see page 5).

Revised ‘Results’ section (see page 5)

“Over half of the respondents (61.9%) had a monthly income of less than 2,500 RMB (i.e., < $386.8); only 14.9% had a monthly income of over 5,000 RMB (i.e., > $773.5).”

Discussion

  1. It is interesting that housing a key predictor of QOL – but discussion says this is inconsistent with previous research. I disagree with this, there is a wealth of evidence and existing literature that housing impacts on QOL. If this statements relates to china alone then this needs clarifying, but again I’d be surprised if there was no evidence for this in Chinese context.

Reply:

Revised ‘Discussion’ section (see page 8)

“Among all AFRC components, only housing exerts the same impact on both the physical and psychological QoL of both middle-aged and older adults, which is inconsistent with previous studies46.  The study conducted in Canada was found out that housing only ex-erted positive impact on health of older adults, but had insignificant relation with health and life satisfaction of younger adults [46].”

  1. Some of the statements are very forthright and need context and toning down, such as “social participation only influences the social QOL of older adults.” – in this study, which used purposive sampling. There will likely be some groups in middle aged people for who social participation influences of social QOL.

 Reply:

Revised ‘Discussion’ section (see page 8)

“Social participation might influence the social QoL of older adults.”

“Outdoor spaces are found out to influence the environmental QoL of older adults, and affect the physical and social QoL of middle-aged people.”

Recommendations

  1. “Public facilities should be installed at the center of rural communities” – such as what?

Reply:

Revised ‘Recommendations’ section (see page 10)

“Public facilities such as physical exercising facilities and billboards should be installed at the center of rural communities so that they are easily accessible for older adults.”

  1. Limitations are well considered- but the authors should also add this was not a randomised sample, and there may have been bias in recruitment and therefore results. That seems like a major limitation.

Revised ‘Recommendations’ section (see page 10)

“Purposive sampling method might induce bias of results. Therefore, it is suggested to adopt random sample in further studies.”

Reviewer 2 Report

Dear authors, 

I would like to congratulate you about this excellent work,  I really enjoyed the reading and the remarkable conclusions. 

I only have few considerations to ask: 

When, into the literature review authors describe age-friendly communities, areas, policies... it could be a point to consider a saving plan of repeated words, to make the readiness easier.

Research methods: Sample size calculation is missing, and also the statistical program and its version. 

Results: TABLES: it could be convenient to bold the significant data in order to make it easier for readers. 

Thank you in advance

Best

Author Response

  1. When, into the literature review authors describe age-friendly communities, areas, policies... it could be a point to consider a saving plan of repeated words, to make the readiness easier.

Reply:

Please note that we use abbreviation of age-friendly communities (i.e., AFC) in the manuscript.

  1. Research methods: Sample size calculation is missing, and also the statistical program and its version. 

Reply:

Revised ‘Research Methods’ section (see page 4)

“The surveys were distributed in 10 rural communities of different regions in China in or-der to mix different background of rural areas. The total aging population in these study regions is near 3,000,000. According to sample size calculation methods, the sample size needs to achieve 385 or more in order to have a confidence level of 95%. Ultimately, 863 valid questionnaires were collected.  Of these, 393 respondents were in the 45–60 age range, which was designated as representing middle-aged people; 470 respondents were aged over 60 and treated as older adults [WHO, 2021; Yu et al., 2019].”

“Multiple statistical methods were adopted, including a Chi-squared test, reliability test, t-test, and linear regression models. The analysis was conducted using SPSS version 24.0.”

  1. Results: TABLES: it could be convenient to bold the significant data in order to make it easier for readers. 

Reply:

Please note that significant data have been bold in tables.

Round 2

Reviewer 1 Report

Thank you for asking me to review this resubmission. This version is improved, and I note that the authors have taken into account some of the reviewers comments - but not all.  Some examples where I think more attention is needed are as follows:

- The authors have not critically engaged with the point made by the reviewer that there is a lot of evidence for housing impacting on QoL / wider wellbeing. The authors are wrong to say the relationship between housing and QoL is inconsistent - at least in European / North American countries there is a well established link.

- I'm also still unclear what information and communication refers to - this really does need clarifying in a Chinese context. An example of this is how the authors helpfully give an example of what public facilities are - and they offer the example of billboards - this is so culturally contextual and not providing examples for other things risks a great deal of misunderstanding among non-Chinese readers.

- The authors also need to state (and not just reference) why the over 60s are deemed to be 'aged' - in most other national / cultural contexts this would not be appropriate or at least deemed very arbitrary. 

Author Response

  1. The authors have not critically engaged with the point made by the reviewer that there is a lot of evidence for housing impacting on QoL / wider wellbeing. The authors are wrong to say the relationship between housing and QoL is inconsistent - at least in European / North American countries there is a well established link.

Reply:

Revised ‘Discussion’ section (see page 8).

“Among all AFRC components, only housing exerts the same impact on both the physical and psychological QoL of both middle-aged and older adults, which is consistent with previous studies46. Housing is one of the key factors for improving quality of life and health of older adults.”

Please kindly note that the original reference has been replaced.

  1. I'm also still unclear what information and communication refers to - this really does need clarifying in a Chinese context. An example of this is how the authors helpfully give an example of what public facilities are - and they offer the example of billboards - this is so culturally contextual and not providing examples for other things risks a great deal of misunderstanding among non-Chinese readers.

Reply:

Revised ‘Literature Review’ section (see page 3)

“Older adults also need to communicate with others and obtain accurate information. In rural communities of China, broadcast and billboards were major approaches to deliver information. Communications systems (e.g., broadcasts and billboards) should be availa-ble to reach community residents of all ages in order to provide up-to-date and available information.  Important notifications and messages are posted on the centrally located billboards for rural community-dwelling residents to easily access necessary infor-mation33.”

  1. The authors also need to state (and not just reference) why the over 60s are deemed to be 'aged' - in most other national / cultural contexts this would not be appropriate or at least deemed very arbitrary. 

Reply:

Please kindly note that current sample was classified as middle-aged people (aged 45-60) and older adults (over 60).

Revised ‘Research methods’ section (see page 4)

“Of these, 393 respondents were in the 45–60 age range, which was designated as representing middle-aged people; 470 respondents were aged over 60 and treated as older adults 37, 43. According to WHO, people aged 60 year and older are counted as aging population44. Hence, respondents over 60 years old are classified as older adults.”
